# Direct real-time RT-PCR for the detection of dengue virus from patient serum in Lao PDR

Vilayouth Phimolsannousith[1], Malavanh Vongsouvath[1], Padthana Kiedsathid[1], Manivanh Vongsouvath[1], Elizabeth A. Ashley[1,2], Audrey Dubot-Pérès[1,2,3]*

1 Lao-Oxford-Mahosot Hospital-Wellcome Trust Research Unit (LOMWRU), Microbiology Laboratory, Mahosot Hospital, Vientiane, Laos PDR, 2 Centre for Tropical Medicine and Global Health, Nuffield Department of Clinical Medicine, University of Oxford, Oxford, United Kingdom, 3 Unite des Virus Emergents (UVE: Aix-Marseille Univ, Universita di Corsica, IRD 190, Inserm, IRBA), Corsica, Marseille, France

* audrey@tropmedres.ac

## Abstract

### Introduction

Dengue fever is a growing global concern with an estimated 100–400 million infections every year and rising mortality over the past decade. In 2017, 40,000 deaths were attributed to dengue. Real-time reverse transcription PCR (RT-qPCR) is the gold standard technique to detect dengue virus (DENV) during the acute phase of the infection. However, it requires prior RNA purification which is costly and time consuming. We evaluated direct RT-qPCR using Luna Universal Probe One-Step RT-qPCR kit (Luna RT-qPCR) for the detection of DENV in sera.

### Methods

Luna RT-qPCR conditions were optimized using DENV2 isolates. The efficiency of direct Luna RT-qPCR was evaluated on a panel of 132 patient sera using RNA purification (EZ1&2 Virus Mini Kit) followed by RT-qPCR (SuperScript III Platinum One-Step qRT-PCR system) as reference standard.

### Results

The sensitivity (95% CI) of direct Luna RT-qPCR using neat patient sera was 34% (25–45). By reducing PCR inhibitors through a 1/10 dilution of the sera, the sensitivity improved to 86% (95% CI: 77–92). Comparable results were obtained between direct Luna RT-qPCR and reference standard process for samples with Cq < 35.

### Conclusion

The results obtained in this study are promising. Direct RT-qPCR for DENV detection in patient sera, could make PCR-based dengue detection and typing, and potentially

**Data availability statement:** All relevant data are within the manuscript and its Supporting Information files.

**Funding:** This study was funded by the Institute of Research for Development (IRD), Aix-Marseille University, and by the Wellcome Trust [Grant number 220211/Z/20/Z]. For the purpose of open access, the author has applied a CC BY public copyright licence to any Author Accepted Manuscript version arising from this submission. The funders had no role in study design, data collection and analysis, decision to publish, or preparation of the manuscript.

**Competing interests:** The authors have declared that no competing interests exist.

other target detections, more affordable for reference laboratories in LMICs by reducing reagent cost by approximately two-thirds. Further studies are needed to evaluate DENV direct RT-qPCR on prospective samples in diagnosis context and to improve the sensitivity by minimizing the impact of inhibitors.

## Introduction

Dengue is a mosquito-borne viral disease that has a major socio-economic and health impact in tropical and subtropical regions. It is caused by one of the four antigenically related serotypes of the dengue virus (DENV). Dengue fever is a growing global concern, with mortality rates rising steadily over the last decade and estimated at more than 40,000 in 2017 [1]. The infection rate has increased 30-fold in the last 50 years. The WHO estimates that 100–400 million infections occur each year [2] and that the disease is spreading to new countries, including temperate regions [3]. More than 50% of the world's population live in dengue-endemic areas, in over 100 countries, and more than 70% (1.8 billion) of the world's population at risk for dengue live in the South-East Asia and Western Pacific regions [4]. Dengue is known as an urban disease, but there is increasing evidence of epidemics in rural areas too [5,6].

In Laos, the first cases of dengue were reported in 1979, and national surveillance based on clinical case reporting was introduced in 2006. Epidemiological data reported over the last decade show that dengue is a growing cause of morbidity and mortality in both urban and rural areas, with epidemics occurring each year during the rainy season and 3.9 million people at risk of DENV infection [5,7–9]. In Laos, the four serotypes of dengue circulate alternately over the years, with the predominant serotype driving the epidemic [10–13]

Infection with DENV is difficult to distinguish clinically from infection with other pathogens circulating in the same region, which is why its confirmatory diagnosis relies on laboratory tests.

Real-time reverse transcription PCR (RT-qPCR) is the gold standard technique for detecting the four DENV serotypes (DENV-1 to −4) during the acute phase of infection, with high sensitivity (95%) and specificity (89%) [14]. Compared with the detection of NS1 antigen, RT- PCR has high analytical specificity, with no cross-reactivity with other flaviviruses, and can distinguish between the four serotypes or to detect co-infection with several serotypes. While detection of NS1 is less sensitive for detecting secondary infection, RT-PCR maintains consistent performance [15]. However, this technique requires a preliminary step of RNA purification from patient samples, which is time-consuming (from 1 to a few hours), costly (~10 euros per sample), may require the use of expensive specific equipment and may be a source of potential cross-contamination.

During the COVID-19 pandemic, given the need for rapid, reliable and cost-effective techniques for the detection of SARS-CoV-2, several studies were conducted to evaluate SARS-CoV-2 RT-qPCR performed directly on biological samples (nasopharyngeal swab in viral transport medium (VTM)) without prior RNA

purification. Compared with the standard process (RNA purification prior to RT-qPCR), the sensitivity of direct RT-qPCR ranged from 76% to 100% [16–22].

For other RNA viruses, only a few studies have been published on the evaluation of direct RT-PCR on clinical samples. As early as 1992, Ravaggi et al. showed that direct RT followed by PCR could detect HCV RNA in serial serum samples from the same patient [23]. In 1994, Morita et al. successfully detected DENV RNA by RT followed by PCR directly from serum samples of three dengue patients [24]. Nishimura et al. validated the replacement of RNA purification by the use of a simple RNA release step for the detection of norovirus RNA by RT-PCR from 275 fecal samples, which showed 94% sensitivity and 95% specificity compared with RT-PCR from RNA purified from the same samples [25]. An evaluation of 126 stored plasmas by direct RT-qPCR using direct blood RT-PCR kit (VitaNavi) showed a sensitivity of 77% and a specificity of 94% for the detection of DENV compared with QIAamp viral RNA purification followed by the RealStar Dengue RT-PCR kit [26].

In the present study we sought to optimize and evaluate direct RT-qPCR using the Luna® Universal Probe One-Step RT-qPCR kit (New England Biolabs) for the detection of DENV in patient serum. Conditions for direct RT-qPCR were first optimized using a DENV strain isolated in cell culture. Then a panel of known dengue-positive and dengue-negative sera from patients admitted to Mahosot hospital, Vientiane, Laos, was used to compare direct RT-qPCR results with the reference process used at Mahosot hospital for dengue diagnosis, consisting of RNA purification followed by RT-qPCR using the SuperScript III Platinum One-Step qRT-PCR system (ThermoFisher).

## Materials and methods

### Ethics statement

Patient sera used in this study were retrospective samples collected as part of the study "A prospective study of the causes of fever amongst patients admitted to Mahosot hospital, Vientiane, Lao PDR" conducted by LOMWRU at Mahosot hospital, Vientiane, Laos, which includes dengue investigation. All patients gave written informed consent to participate in the study. Ethical approvals were provided by the Lao National Ethics Committee for Health Research (NECHR 02, NECHR 028; NECHR 049; NECHR 046), the University of Health Science Research Ethics Committee (113/REC, 158/REC, 285/REC) and the Oxford Tropical Ethics Committee (OXTREC 006−07, OXTREC 041−20). Data were accessed for research purposes on 2nd March 2022. The authors didn't have access to information identifying individual participants.

### Preparation of DENV2 isolate dilutions

DENV2 had previously been isolated in a BSL3 (biosafety level 3) laboratory by inoculating cell culture with dengue patient serum. The DENV2 cell culture supernatant (DENV2 isolate) was aliquoted and stored at −80°C. Tenfold serial dilutions of DENV2 isolate ($10^{-1}$ to $10^{-8}$) were prepared using RPMI medium with 5% Fetal Bovine Serum (FBS). Four aliquots of 200 µL each were prepared for each dilution and frozen at −80°C until use. Aliquots were used immediately after thawing and never refrozen.

### Patient sera

As part of the study "A prospective study of the causes of fever amongst patients admitted to Mahosot Hospital, Vientiane, Lao PDR", blood samples were taken from patients admitted to Mahosot hospital with clinical suspicion of dengue fever according to WHO criteria [27]. Sera were submitted to routine dengue diagnosis at the Microbiology Laboratory, including RT-qPCRs for DENV detection and serotyping. Serum aliquots were stored at −80°C immediately after blood collection. For the present study, a panel of samples from 2013 to 2020 was selected on the basis of positive DENV RT-qPCR results: 29 sera from patients with DENV1, 27 with DENV2, 22 with DENV3, and 24 with DENV4, a third with a RT-qPCR Cq value of 30–40, a third with a Cq value of 20–30, and a third with a Cq value below 20. Thirty DENV RT-qPCR negative sera were also included.

### RNA purification for the reference standard process

In BSL3, an aliquot of each dilution of the DENV2 isolate (8 dilutions from $10^{-1}$ to $10^{-8}$) was extracted using the QiAamp® viral RNA kit (Qiagen) following the manufacturer's instructions using 140 μL of sample and 80 μL of elution volume.

RNA was purified from 100 μL of frozen patient serum using the EZ1&2™ Virus Mini Kit v2.0 (Qiagen) following the manufacturer's instructions and 60 μL elution volume.

RNA samples from serum and DENV2 isolate were stored at −80°C until RT-qPCR testing.

### Real-time RT-PCR (RT-qPCR)

**Primers and probe.** DENV RNA was detected using a pan-DENV RT-qPCR system described previously [28], allowing amplification of all four DENV serotypes.

In this study, the primers and probe were used in two forms: 1) individual primers and probe regenerated at 10 μM (liquid primes and probe), 2) a pre-prepared lyophilized mixture of primers and probe (Lyoph P&P) [29]. For the reference standard RT-qPCR, Lyoph P&P were resuspended, just before use, using 357 μL of AE buffer for vials prepared for 51 reaction, or 182 μL for vials prepared for 26 reactions. For the Direct RT-qPCR, the Lyoph P&P were resuspended, just before use, using 137 μL of AE buffer for vials prepared for 51 reaction, or 70 μL for vials prepared for 26 reactions.

**Reference Standard RT-qPCR process following RNA extraction.** The RT-qPCR was performed using the SuperScript III Platinum One-Step RT-qPCR kit (ThermoFisher, SuperScript III RT-qPCR kit) following the method described previously [28]. Briefly, RT-qPCR was performed using a 25 μL RT-qPCR mix containing 12.5 μL of 2x Reaction mix, 0.5 μL of SuperScript™ III RT/Platinum™ Taq Mix, 400 nM of each reverse and forward primer, and 160 nM of probe (either 1 μL of each 10 μM primer and 0.4 μL of 10 μM probe; or 7 μL of regenerated Lyoph P&P vial) [29], and 5 μL of purified RNA. The RT-qPCR cycling conditions used were: 50°C for 15 minutes, 95°C for 2 minutes, followed by 45 cycles of 95°C for 15 seconds and 60°C for 45 seconds.

**Direct RT-qPCR using Luna RT-qPCR Kit.** Luna® Universal Probe One-Step RT-qPCR kit (NEB, Luna RT-qPCR kit) was used following the protocol developed at UVE (Unit des Virus Emergents, Marseille, personal communication from G. Piorkowski) using a 14 μL RT-qPCR mix containing 5 μL of Luna Universal Probe One-Step Reaction Mix (2X), 0.5 μL of Luna WarmStart RT Enzyme Mix (20X), 400 nM of each forward and reverse primer and 200 nM of probe (either 0.6 μL of each 10 μM primer and 0.3 μL of 10 μM probe, or 1.5 μL of regenerated Lyoph P&P) and 7 μL of sample. Sample (serial dilutions of DENV2 isolate or patient sera) were use either neat or diluted 1/10 in molecular grade water.

Using liquid primer and probe solution, several primer concentrations were evaluated (200 nM, 600 nM and 800 nM); an increase in the concentration of primers and probes (400 nM/200 nM to 600 nM/300 nM) was also tested using regenerated Lyoph P&P.

The RT-qPCR cycling conditions used were: 55°C for 10 min, then 95°C for 1 minute, followed by 40 cycles of 95°C for 10 seconds, and 60°C for 60 seconds. We also tested 56°C and 58°C as annealing temperatures, so the cycling was set as following: 55°C for 10 min, then 95°C for 1 min, followed by 40 cycles 95°C for 10sec, 56°C or 58°C for 15 seconds, and 60°C for 45 seconds.

Reference Standard RT-qPCR and direct RT-qPCR were both performed using the CFX96 Touch Real-Time PCR Detection System (Bio-Rad) with synthetic RNA control [29] (each positive control tube was used no more than a few times) and no template negative control was added in each run.

### Result interpretation and Analysis

Positive RT-qPCR result was defined by a sigmoidal shape amplification curve with a Cq value < 40, and negative result was defined as no Cq value or Cq value ≥ 40.

The limit of detection (LOD) was assessed, defined as the last dilution for which a positive RT-qPCR result was obtained for both duplicates.

Accuracy, sensitivity and specificity of direct RT-qPCR to detect DENV were calculated by comparison with results obtained by the reference standard process (RNA purification followed by SuperScript III RT-qPCR). Direct RT-qPCR and the reference standard process were compared by calculating Cohen's Kappa coefficient. These analyses were also performed with RT-qPCR results obtained using Cq < 36 as the positivity cutoff.

## Results

### Optimisation of direct RT-qPCR using DENV2 isolate

**Comparison of direct RT-qPCR with the reference standard RT-qPCR process.** Dilutions $10^{-1}$ to $10^{-8}$ of DENV2 isolate were submitted to direct RT-qPCR using Luna RT-qPCR kit and to the reference standard RT-qPCR process consisting of RNA purification followed by RT-qPCR using SuperScript III RT-qPCR kit. The reference standard RT-qPCR process showed a better LOD ($> 10^{-8}$) compared to direct RT-qPCR, which showed a LOD of $10^{-6}$ (Table 1).

**Evaluation of RT-qPCR inhibitors.** To determinate whether any inhibitors present in the DENV2 isolate samples would have interfered with Luna RT-qPCR, each dilution of DENV2 isolate samples was tested neat and diluted 1/10 in water by Luna RT-qPCR; duplicates were performed for all. The results show (S1 Table) higher Cq values for the 1/10 diluted samples than for the undiluted samples (with an average difference of 4), indicating the absence of RT-qPCR inhibitors.

**Optimization of primer and probe concentration.** In order to improve the LOD of direct Luna RT-qPCR, we tested different primers and probes concentrations: 1) liquid primers at concentrations of 200 nM, 400 nM, 600 nM and 800 nM; 2) Lyoph P&P at concentrations of 400 nM/200 nM and 600 nM/300 nM. A LOD of $10^{-6}$ was observed for RT-qPCR performed using liquid primers at concentrations 200 nM, 400 nM or 800 nM (Table 2). In contrast, a LOD of $10^{-7}$ was observed for RT-qPCR performed using liquid primers at concentration 600 nM or Lyoph P&P at both concentrations tested.

**Table 1. Results of Luna RT-qPCR performed directly on DENV2 isolate dilutions and of SuperScript III RT-qPCR performed on the same samples after RNA purification.**

| Template | Direct Luna RT-qPCR (Cq) | | Reference standard RT-qPCR (Cq) | |
|---|---|---|---|---|
| | Without RNA extraction | | with RNA extraction | |
| DENV2 $10^{-1}$ | 19.01 | 19.65 | 17.41 | 17.70 |
| DENV2 $10^{-2}$ | 22.12 | 21.79 | 21.01 | 20.86 |
| DENV2 $10^{-3}$ | 25.78 | 25.50 | 25.78 | 25.61 |
| DENV2 $10^{-4}$ | 28.95 | 29.91 | 29.58 | 29.54 |
| DENV2 $10^{-5}$ | 32.32 | 32.50 | 33.16 | 32.89 |
| DENV2 $10^{-6}$ | 37.27 | 35.01 | 34.51 | 34.63 |
| DENV2 $10^{-7}$ | neg | neg | 36.67 | 36.55 |
| DENV2 $10^{-8}$ | neg | neg | 37.94 | 39.08 |
| Pos (high) | 21.44 | 21.45 | 22.44 | 22.33 |
| Pos (medium) | 24.98 | 24.78 | 25.69 | 26.00 |
| Pos (low) | 27.95 | 27.80 | 28.66 | 28.51 |

Templates were from 8 serials dilutions ($10^{-1}$ to $10^{-8}$) of DENV2 isolate, directly used for Luna RT-qPCR (without sample dilution) or submitted to RNA purification for reference standard process. Cq = quantification cycle; neg = negative result, no Cq. Pos = synthetic RNA in three concentrations (high, medium and low) was used as RT-qPCR positive control. Grey shade: last dilution with both duplicates with positive RT-qPCR result (Cq < 40), corresponding to the limit of detection (LOD). Liquid primers (400nM) and liquid probe (200nM) were used for Luna RT-qPCR.

**Table 2. Results of Luna RT-qPCR performed on serial dilutions of DENV2 isolate using liquid primers at different concentrations, and pre-prepared freeze-dried mix of primers and probe at two concentrations.**

| Template | Direct Luna RT-qPCR (Cq) | | | | | | | | | | | |
| --- | --- | --- | --- | --- | --- | --- | --- | --- | --- | --- | --- | --- |
| | Using liquid primer and probe (200 nM**) | | | | | | | | Using Lyoph P&P | | | |
| | 200nM* | | 400nM* | | 600nM* | | 800nM* | | 400nM*/200nM** | | 600nM*/300nM** | |
| DENV2 $10^{-4}$ | 30.79 | 29.69 | 28.95 | 29.91 | 29.53 | 30.37 | 28.88 | 28.90 | 26.74 | 26.51 | 25.85 | 25.79 |
| DENV2 $10^{-5}$ | 34.26 | 33.88 | 32.32 | 32.50 | 34.26 | 33.07 | 32.10 | 32.12 | 30.07 | 29.58 | 29.81 | 29.61 |
| DENV2 $10^{-6}$ | 35.95 | 38.05 | 37.27 | 35.01 | 35.85 | 35.64 | 36.23 | 35.44 | 33.94 | 33.45 | 33.19 | 33.43 |
| DENV2 $10^{-7}$ | neg | neg | neg | neg | 37.65 | 36.78 | neg | neg | 37.20 | 37.30 | 35.86 | 37.78 |
| DENV2 $10^{-8}$ | neg | neg | neg | neg | neg | neg | neg | neg | 39.14 | No Cq | 39.59 | No Cq |
| Pos (high) | 21.78 | 21.94 | 21.44 | 21.45 | 22.01 | 22.08 | 22.29 | 22.19 | 23.51 | 23.38 | 23.62 | 23.91 |
| Pos (med) | 25.12 | 24.99 | 24.98 | 24.78 | 24.78 | 24.76 | 25.39 | 25.87 | 26.63 | 26.99 | 26.69 | 27.00 |
| Pos (low) | 28.04 | 28.39 | 27.95 | 27.80 | 28.24 | 29.86 | 29.01 | 29.03 | 30.54 | 30.45 | 30.11 | 30.13 |

Lyoph P&P = pre-prepared freeze-dried mix of primers and probe. RT-qPCR templates were serial dilutions ($10^{-4}$ to $10^{-8}$) of DENV2 isolate without prior RNA purification. Cq = quantification cycle. Neg = negative result, no Cq; Pos = synthetic RNA in three concentrations (high, medium and low) used as RT-qPCR positive control. * = concentration of primers in the RT-qPCR mix. ** = concentration of the probe in the RT-qPCR mix. Grey shade: last dilution with both duplicate with positive RT-qPCR result (Cq < 40), corresponding to the limit of detection (LOD).

**Optimization of annealing temperature.** Three annealing temperatures (56°C, 58°C and 60°C) for direct Luna RT-qPCR were tested on serial dilutions ($10^{-6}$ to $10^{-10}$) of DENV2 isolate using Lyoph P&P at the 400nM/200 nM concentration (giving the best LOD with the lowest primer concentration). The best LOD ($10^{-8}$) was obtained at an anneling temperature of 56°C (Table 3).

## Comparison of direct RT-qPCR with the reference standard RT-qPCR process for DENV detection in serum samples

A panel of 132 patient sera was tested, undiluted and diluted 1/10, by direct Luna RT-qPCR using the optimized protocol above (Lyoph P&P at 400 nM/200 nM concentration, and 56°C as annealing temperature). The same sera were then (after one freeze/thaw cycle) submitted to reference standard RT-qPCR process consisting of RNA purification followed

**Table 3. Results of Luna RT-qPCR using freeze-dried primers and probes and three annealing temperatures on dilutions of DENV2 isolate.**

| Template | Direct Luna RT-qPCR (Cq) | | | | | |
| --- | --- | --- | --- | --- | --- | --- |
| | Annealing T° 56°C | | Annealing T° 58°C | | Annealing T° 60°C | |
| DENV2 $10^{-6}$ | 32.50 | 32.72 | 33.14 | 32.50 | 32.32 | 30.73 |
| DENV2 $10^{-7}$ | 36.78 | 35.61 | 34.07 | 36.08 | 34.93 | 34.96 |
| DENV2 $10^{-8}$ | 38.30 | 39.45 | neg | neg | 36.82 | neg |
| DENV2 $10^{-9}$ | neg | 39.26 | neg | neg | neg | neg |
| DENV2 $10^{-10}$ | neg | neg | neg | neg | neg | neg |
| Pos (high) | 23.33 | 23.30 | 22.76 | ND | 22.33 | ND |
| Pos (medium) | 26.41 | 25.95 | 26.28 | ND | 26.14 | ND |
| Pos (low) | 29.95 | 29.82 | 29.11 | ND | 28.35 | ND |

RT-qPCR templates were serial dilutions ($10^{-6}$ to $10^{-10}$) of DENV2 isolate without prior RNA purification. Cq = quantification cycle. Neg = negative result, no Cq. ND = not done. Pos = synthetic RNA in three concentrations (high, medium and low) used as RT-qPCR positive control. Grey shade: last dilution with both duplicates with positive RT-qPCR result (Cq < 40), corresponding to the limit of detection (LOD). Pre-prepared freeze-dried mix of primers and probe (Lyoph P&P) at concentration 400 nM/200 nM was used.

by SuperScript III RT-qPCR. The Cq values obtained for all RT-qPCRs are presented in supplementary data (S2 Table). Table 4 compares the results obtained by direct Luna RT-qPCR with those obtained by the reference standard RT-qPCR process. Of the 93 samples found to be positive by the reference standard RT-qPCR process, only 32 (34%) were also positive by direct Luna RT-qPCR performed on undiluted sera, but 80 (86%) were positive by direct Luna RT-qPCR on 1/10 diluted sera. When the cutoff for positive RT-qPCR results was set at Cq < 36, direct Luna RT-qPCR on 1/10 diluted sera achieved 100% sensitivity. The Cq values showed no significant difference for DENV2 (Wilcoxon matched-pairs signed-rank test p = 0.24) and DENV3 (p = 0.18). However, for DENV1 and DENV4, Cq values were significantly higher for direct Luna RT-qPCR compared with the reference standard RT-qPCR process (p < 0.001) (S2 Table, Fig 1, Fig 2).

## Discussion

In this study we were able to detect DENV directly from cell culture isolates and patient serum using Luna® Universal Probe One-Step RT-qPCR kit (NEB) without prior RNA purification. Nowadays a wide range of commercial kits are available for RT-qPCR, they are often presented as having been developed for a dedicated use. A few RT-PCR kits are marketed specifically for direct RT-qPCR without prior RNA purification, but the few kits that were evaluated in preliminary trials at the UVE did not give promising results (personal communication from G. Piorkowski). In the current study, a two-log improvement in the analytical sensitivity of direct Luna RT-qPCR was observed by increasing the primer concentration and decreasing the annealing temperature, giving comparable results for direct RT-qPCR and reference RT-qPCR with prior RNA purification, when testing serial dilutions of DENV2 isolate.

The accuracy of the optimized direct Luna RT-qPCR was assessed against the reference standard (RNA purification followed by SSIII RT-qPCR) on a panel of 132 patient sera. A low sensitivity of 34%, was observed. The samples with the highest Cq values by the reference standard (≥31 for DENV1, ≥ 28 for DENV2, ≥ 27 for DENV3 and ≥24 for DENV4) were not detected by direct Luna RT-qPCR (S2 Table). This loss of efficiency when testing sera in comparison to the viral

**Table 4. Agreement between results of Luna RT-qPCR and reference standard RT-qPCR process using non-diluted and diluted patient sera.**

| Positive RT-qPCR result was determined as Cq value < 40 | | | | | Positive RT-qPCR result was determined as Cq value < 36 | | | | |
|---|---|---|---|---|---|---|---|---|---|
| | | Reference standard RT-qPCR process | | Total | | | Reference standard RT-qPCR process | | Total |
| | | Pos | Neg | | | | Pos | Neg | |
| **Luna RT-qPCR on non-diluted samples** | Pos | 32 | 0 | 32 | **Luna RT-qPCR on non-diluted samples** | Pos | 32 | 0 | 32 |
| | Neg | 61 | 39 | 100 | | Neg | 39 | 61 | 100 |
| **Total** | | 93 | 39 | 132 | **Total** | | 71 | 61 | 132 |
| Sensitivity (95% CI): 34.4% (24.9–45.0) | | | | | Sensitivity (95% CI): 45.1% (33.2–57.3) | | | | |
| Specificity (95% CI): 100% (91.0–100) | | | | | Specificity (95% CI): 100% (94.1–100) | | | | |
| Accuracy (95% CI): 53.8% (44.9–62.5) | | | | | Accuracy (95% CI): 70.5% (61.9–78.1) | | | | |
| Cohen's kappa coefficient: 0.23 | | | | | Cohen's kappa coefficient: 0.43 | | | | |
| **Luna RT-qPCR on diluted samples** | Pos | 80 | 0 | 80 | **Luna RT-qPCR on diluted samples** | Pos | 71 | 3 | 74 |
| | Neg | 13 | 39 | 52 | | Neg | 0 | 58 | 58 |
| **Total** | | 93 | 39 | 132 | **Total** | | 71 | 61 | 132 |
| Sensitivity (95% CI): 86.0% (77.3–92.3) | | | | | Sensitivity (95% CI): 100% (94.9–100) | | | | |
| Specificity (95% CI): 100% (91.0–100) | | | | | Specificity (95% CI): 95.1% (86.3–99.0) | | | | |
| Accuracy (95% CI): 90.2% (83.7–94.7) | | | | | Accuracy (95% CI): 97.7% (93.5–99.5) | | | | |
| Cohen's kappa coefficient: 0.78 | | | | | Cohen's kappa coefficient: 0.92 | | | | |

Pos:positive, Neg: negative. Reference standard RT-qPCR process: SuperScript III RT-qPCR performed on purified RNA.

**Fig 1. Comparison of Cq values obtained for a same sample by reference standard process (RNA purification followed by SSIII RT-qPCR) and direct Luna RT-qPCR on 1/10 diluted serum.** Negative results with no Cq value are represented in the figure by Cq of 45.

isolate was due to the presence of PCR inhibitor. This result was confirmed by performing direct RT-qPCR on diluted sera, which permitted to increase the sensitivity to 86%. The sensitivity calculated for each serotype ranged from 75% for DENV2–100% for DENV1 (S3 Table). However, the 95% confidence intervals were wide, due to the low number of samples tested for each serotype; in addition the highest Cq value for DENV1 was 37, while it was 39 for the other serotypes. For all serotypes, the direct RT-qPCR gave good results for all samples that had a Cq<36 with the reference standard. However, for DENV1 and DENV4, significantly higher Cq values were observed with direct Luna RT-qPCR compared to the reference standard process (Wilcoxon matched-pairs signed-rank test p<0.001). For all serotypes, there was a clear loss of sensitivity for samples with Cq higher than 35.

For the reference standard process, the PCR mix contains the amount of viral RNA present in 8 µL of serum (100 µL of serum eluted in 60 µL then 5 µL of RNA used in PCR mix). For direct RT-qPCR, the PCR mix contains the amount of viral RNA present in 0.7 µL of serum (7 µL of serum diluted 1/10). Ten times less RNA template was used in the direct RT-qPCR compared with the reference standard RT-qPCR, which should correspond to a difference of 3 Cq. However, for samples with Cq ≤ 35, the average difference in Cq value between the reference standard and direct RT-qPCR is 0.6. This suggests that Luna RT-qPCR kit would be more efficient than the SSIII RT-PCR kit. The direct RT-qPCR could potentially be improved by adding a chelator to the PCR mastermix to reduce the effect of PCR inhibitor present in the serum, so the serum dilution would not be necessary and more RNA template would be available in the PCR mix.

Only one published study has evaluated the use of direct RT-PCR for the detection of DENV on a panel of patient samples [26]. The performance of direct RT-qPCR using the Direct Blood RT-PCR kit (VitaNavi Technology) was evaluated on a panel of 126 patient plasmas, using RNA purification followed by RT-qPCR using the RealStar Dengue PCR kit (2.0) (Altona Diagnostics) as reference standard. Sensitivity (76.7%) and specificity (93.9%) were slightly lower than in our

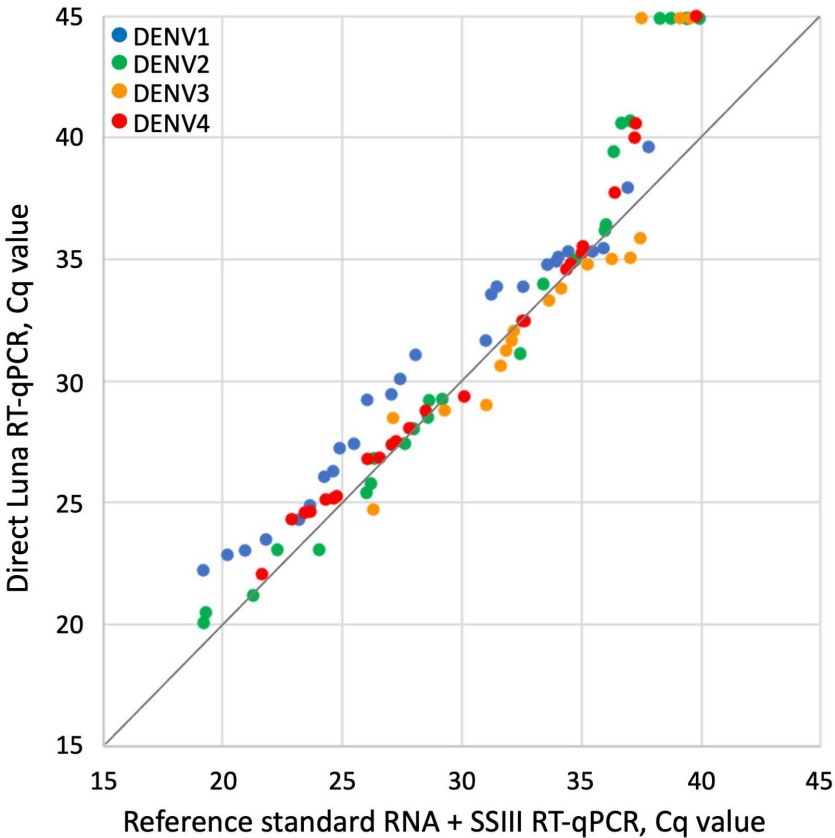

**Fig 2. Comparison of Cq values obtained with reference process (RNA purification followed by SSIII RT-qPCR) and direct Luna RT-qPCR on 1/10 diluted serum.** DENV1 samples are in blue, DENV2 samples in green, DENV3 samples in orange and DENV4 samples in red. Negative results with no Cq value are represented in the figure by Cq of 45.

study. They used SybrGreen RT-qPCR whereas we used probe-based RT-qPCR, which is a more specific real-time PCR technique. They used plasma at a final dilution of 1/10 final in the PCR mix, whereas for our study we used serum to a final dilution of 1/20 in the PCR mix. Consequently, a lower dilution of the PCR inhibitor in the plasma could result in lower sensitivity.

Several articles have been published on the use of direct RT-qPCR for the detection of SARS-CoV-2 [16–22]. Overall, better sensitivities were observed compared with our study (from 81% to 100%). These differences in sensitivity can be explained in various ways, as we will discuss below.

The first factor to consider is the template used for direct RT-qPCR. Indeed, SARS-CoV-2 direct detection was achieved using VTM which contains fewer PCR inhibitors than serum.

Another important factor is the characteristics of the panel of samples used to evaluate the tests. In our study, we selected samples to have a wide range of Cq values, so it is possible that the proportion of samples with high Cq is higher than in the SARS-CoV-2 studies. Indeed, Barza *et al.*, observed, as in our study, negative results by direct RT-qPCR for samples with Cq > 35 with the reference standard [16]. Lubke *et al.*, observed a sensitivity of 81.3% using a panel of 91 positive samples [18]. Sensitivity increased to 95.8% by selecting only samples with Cq < 35 with the reference standard.

Sample pretreatment could also be an important factor. Some studies have suggested that pre-heating the sample may improve the release of RNA into the PCR mix and therefore the detection of SARS-CoV-2 by dirrect RT-qPCR. However,

while Bruce *et al.* [20] observed a better sensitivity for pre-heated samples (92%) compared with non-preheated samples (84%), Morecchiato *et al.* observed that pre- exposure to heat decreased sensitivity (from 82% to 74%) [22], suggesting that sample pre-heating does not play a major role in improving PCR efficiency.

The differences in reagent used for the direct RT-qpCR would undoubtally have an effect on sensitivity. Smyrlaki *et al.* evaluated different PCR conditions [30]. They showed certains differences in the efficiency of direct RT-qpCR depending on the PCR system (primers and probe) used, a shorter amplicon allowing better efficiency of heat inactivated direct RT-qPCR. Visseaux *et al.*[31], showed important differences in sensitivity depending on the kit used for direct RT-qPCR (55%,70% and 70% when using PrimeDirect, PrimeScript and Sansure assays respectively).

In summary, many factors may play a role in the difference in sensitivity observed in the different studies: the volume of the sample used, the range of Cq values of the sample panel used, the PCR system used, the RT-qPCR kit used, or the nature of the sample used as PCR template.

One of the limitations of this study is the use of retrospective frozen samples with a limited number of samples for each serotype, and we didn't perfom a sample size calculation to justify this number.

Although rapid diagnosis tests for dengue NS1 antigen are unquestionably the most appropriate first-line diagnostic tool in health centres, particularly in remote areas, PCR tests remain the technique of choice for reference laboratories due to their high specificity and sensitivity, as well as their ability to identify circulating serotypes. The results obtained in this study, using Luna RT-qPCR for direct detection of DENV in patient sera, are promising and could considerably reduce the cost of the PCR assay (from around 14 USD to 4 USD per sample), as well as the time to perform the assay (and so the labour cost) by more than twofold (from around 4 hours to less than 2 hours), making it more affordable for diagnosis of dengue in LMICs. The simplification of the assay should not be overlooked either. By halving the number of steps involved in handling biological samples, the risk of contamination is significantly reduced – an important factor in molecular biology laboratories, especially in a context where training technical staff remains a challenge. However, further studies need to be carried out on a larger number of prospective patient samples in order to obtain more precise data on the efficiency of direct Luna RT-qPCR for the diagnosis of dengue in epidemic and endemic contexts. Further experiments should be carried out to improve the efficiency of direct RT-qPCR, such as the use of inhibitor chelator and the evaluation of other PCR systems with different amplicon sizes. Further investigation is needed to extend the use of direct RT-qPCR to other pathogens, enabling it to be applied to a syndromic diagnostic approach. If it proves effective for other targets and sample types, this technique could simplify molecular testing and improve the testing capacity of diagnostic laboratories in LMICs.

## Supporting information

**S1 Table. Results of Luna RT-qPCR performed on neat and 1/10 diluted DENV2 isolate solutions.**
(DOCX)

**S2 Table. RT-qPCR results (Cq values) for direct Luna RT-qPCR and gold standard RT-qPCR process (with prior RNA purification) from patient sera.**
(DOCX)

**S3 Table. Sensitivity of direct Luna RT-qPCR from 1/10 diluted sera for the detection of DENV in comparison to reference standard process (RNA purification followed by SSIII RT-PCR) according to the DENV serotype.**
(DOCX)

## Acknowledgments

We would like to thank all the participants to this study, the Directors and staff of Mahosot Hospital, Paul Newton, the former director of LOMWRU, the staff of the Microbiology Laboratory and LOMWRU for their technical help and supports, and Xavier de Lamballerie and Géraldine Piorkowski at UVE for the support and methodological advice they provided.

## Author contributions

**Conceptualization:** Audrey Dubot-Pérès.

**Data curation:** Audrey Dubot-Pérès.

**Formal analysis:** Vilayouth Phimolsannousith, Audrey Dubot-Pérès.

**Funding acquisition:** Elizabeth A Ashley.

**Investigation:** Vilayouth Phimolsannousith, Malavanh Vongsouvath, Padthana Kiedsathid.

**Project administration:** Elizabeth A Ashley.

**Resources:** Manivanh Vongsouvath, Elizabeth A Ashley, Audrey Dubot-Pérès.

**Supervision:** Manivanh Vongsouvath, Elizabeth A Ashley, Audrey Dubot-Pérès.

**Validation:** Audrey Dubot-Pérès.

**Visualization:** Audrey Dubot-Pérès.

**Writing – original draft:** Vilayouth Phimolsannousith, Audrey Dubot-Pérès.

**Writing – review & editing:** Vilayouth Phimolsannousith, Elizabeth A Ashley, Audrey Dubot-Pérès.

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
