## [Decision Letter · Decision Letter 0]

6 May 2025

Dear Dr. Dubot-Pérès,

Thank you for submitting your manuscript to PLOS ONE. After careful consideration, we feel that it has merit but does not fully meet PLOS ONE’s publication criteria as it currently stands. Therefore, we invite you to submit a revised version of the manuscript that addresses the points raised during the review process.

We look forward to receiving your revised manuscript.

Kind regards,

Nilanka Perera, MD, PhD

Academic Editor

PLOS ONE

Reviewers' comments:

Reviewer's Responses to Questions

**Comments to the Author**

1. Is the manuscript technically sound, and do the data support the conclusions?

Reviewer #1: Yes

Reviewer #2: Partly

2. Has the statistical analysis been performed appropriately and rigorously?

Reviewer #1: Yes

Reviewer #2: Yes

3. Have the authors made all data underlying the findings in their manuscript fully available?

Reviewer #1: Yes

Reviewer #2: Yes

4. Is the manuscript presented in an intelligible fashion and written in standard English?

Reviewer #1: No

Reviewer #2: Yes

Reviewer #1: Line 83: please provide the sensitivity and specificity of RT-PCR with a reference

The introduction is quite lengthy and some of it can be moved to discussion. Given that the audience who would be interested in reading this will anyway be familiar with dengue, there is no need to stress the burden of illness. You can probably get straight in to discussing the diagnostics and their limitations.

Line 201: Positive RTqPCR result was determined as Cq value < 40, and negative was determined as no Cq value or Cq value > 40. The cut-off of 40 is too high. For "positive" results above a Cq of 35 did you run the qPCR product on a gel to make sure that amplification had happened, and that its not a false positive?

Table 2 can be put in to supplementary material

I would not trust even the reference RT-qPCR results for cq values above 35 unless the amplified product is seen on a gel. So your sensitivity and specificity might further improve if do this validation step of samples with high Cq values.

Line 321: What is the PCR inhibitor you are referring to? if its sera it is unlikely this is due to an anticoagulant.

Line 325: Why cannot you test more samples? and how was the sample size calculated?

Line 328 refers to a DENV4 for the first time. I thought there were no DENV4 samples tested?

Lines 331 - 341 : I dont understand what the authors are trying to say here

Lines 345 : If they used RNA purification how can it be direct RT-qPCR?

Lines 347 - 349: So how do you relate the probe and dilution diferences to the differences in results? If you can offer an explanation what is the point of mentioning it as there can be many other differences between the two protocols.

Line 353: What is VTM?

Lines 350 - 379 - This long paragraph has some useful information but its very poorly constructed, lacks a logical flow and eventually disintegrates to a difficult to understand jargon. I suggest splitting it up, signposting your intention throughout (e.g., Several factors contribute to the success rate of direct RT qPCR as evidenced from prior studies including those on other viruses. firstly .....)

The limitations section is very brief - you do not have a sample size calculation, any "postive" result over cq of 35 is questionable, there was no DENV4 - all these are limitations

The conclusions suggest that the new method "drastically" cuts costs - for this conclusion at a bare minimum you must provide a cost-comparison including labour costs. Also it is interesting to know how quickly you can do this test compared to standard RT-qPCR

Overall this manuscript has publishable data. However the quality of written English needs to improve, specially in the discussion.

Reviewer #2: While this research can be useful as groundwork the results provided are not sufficient yet to draw solid conclusions on the use of direct PCR. Would like to suggest broaden the experiment by

*adding dengue virus types other than DENV-2

*including data on cost effectiveness - giving actual amounts compared to other methods of dengue diagnosis

*including data on time saving - giving actual amount of time saved compared to standard methods

**Do you want your identity to be public for this peer review?** For information about this choice, including consent withdrawal, please see our Privacy Policy

Reviewer #1: No

Reviewer #2: No

---

## [Author Response · Author response to Decision Letter 1]

23 Jun 2025

Reviewer #1:

1:

We have provided the sensitivity and specificity with a reference.

2:

Thank you for the comment, the introduction has been shorterned.

3:

A key advantage of real-time RT-PCR is that results can be obtained without ever opening the reaction tube, eliminating the risk of amplicon contamination. For this reason, in our diagnosis laboratory, we never load on a gel the DENV RT-qPCR products. In addition, the real-time RT-PCR being more sensitive than conventionnal RT-PCR, low‑positive samples would not produce visible bands on a gel anyway. We acknowledge that a high Cq values can sometime be hard to distinguish between a low-positive and a false-positive result. However, the use of a hydrolysis probe ensures high specificity of the amplification, which is further visualized by a sigmaoidal curve. Concequently, when we interpret PCR results, we ensure that the signal detected corresponds to an amplification signal represented by a sigmoidal curve. To clarify the interpretation of the result, we have added it to the RT-qPCR positivity criteria.

Line 191: Positive RT-qPCR result was defined by a sigmoidal shape amplification curve with a Cq value < 40

We agree that it is interesting to examine the results when using Cq<36 as the positivity cutoff. We have therefore added these results to Table 5 and indicated this analysis in the method section:

Line 198: These analyses were also performed with RT-qPCR results obtained using Cq < 36 as the positivity cutoff.

We also add one sentence in the text to refer to these results:

Line 270: When the cutoff for positive RT-qPCR results was set at Cq <36, direct Luna RT-qPCR on 1/10 diluted sera achieved 100% sensitivity

4:

Table 2 has been moved to supplementary material.

5:

In table 5 we have added a 2x2 table presenting RT-qPCR results when using Cq<36 as the positivity cuttoff.

6:

We are referring to any serum component that might inhibit PCR (such as immunoglobulins). In our experiment, we don’t know exactly which serum component inhibits the RT-qPCR, but the fact that the Cq value is lower for direct RT-qPCR on 1/10 diluted sera than for RT-qPCR on the same sera tested pure is an evidence of the presence of PCR inhibitor in the pure serum.

7:

We didn’t calculate the sample size, because before testing a larger number of samples, we wanted to optimize the PCR conditions and perform a proof-of-principle experiment for all four DENV serotypes and for a panel of different Cq values, as explained in the sample, selection methods. We didn’t have the resources to test for a larger number of samples for all four serotypes and for a panel of different Cq values. We recognize this as a limitation of our study and have already mentionned the need to test larger a larger number of samples in the discussion (line 384): “further studies need to be carried out on a larger number of prospective patient samples in order to obtain more precise data on the efficiency of direct Luna RT-qPCR for the diagnosis of dengue in epidemic and endemic contexts”, and the fact that this is a limitation of our study (line 370): “One of the limitations of this study is the use of retrospective frozen samples with a limited number of samples for each serotype”.

8:

The study includes patient sera positive for DENV4, as indicated in the mehods section line 134. DENV4 is consistently referenced throughout the results section in relation to patient sera, with corresponding data presented in the figures and supplementary materials.

9:

We want to say that our results suggest that the Luna RT-qPCR kit demonstrates greater efficiency than the SSIII RT-PCR kit. In the direct RT-qPCR, we used ten times less template RNA compared to the SSIII RT-PCR. Based on this difference, we would have expected a Cq value shift of approximately 3 cycles; however the observed difference was less than 1 cycle. This indicates higher efficiency of the Luna RT-qPCR kit. We believe that by increasing the amount of template in the direct RT-qPCR and mitigating PCR inhibition (for ex: through the addition of a chelating agen) we could achieve even greater sensitivity. We hope this explanation is clearer, and we remain happy to rephrase this paragraph if the reviewer deems it necessary.

10:

The RNA purification mentionned in line 332 refers to the reference standard used for comparison with the direct RT-qPCR.

11:

Probe-based real-time PCR is more specific than SybrGreen real-time PCR, which may account for the higher specificity observed. Using 1/20 sample dilution likely reduces the effect of PCR inhibitors compared to 1/10 dilutions, which may account for the higher sensitivity. We have clarified this:

Line 334: They used SybrGreen RT-qPCR whereas we used probe-based RT-qPCR, which is a more specific real-time PCR technique. They used plasma at a final dilution of 1/10 final in the PCR mix, whereas for our study we used serum to a final dilution of 1/20 in the PCR mix. Consequently, a lower dilution of the PCR inhibitor in the plasma could result in lower sensitivity.

12:

VTM stands for viral transport medium. We have added the full term at its first mention on line 81.

13:

Thank you for the comment. We agree that the original paragraph was not always clear. We have revised it by dividing it into several shorter subparagraphs and simplifying certain aspects for better clarity.

14:

The limitation regarding the small sample size has already been acknowledged on line 370. Since we tested DENV4 samples, this does not represent a limitation. Additionnally, we have included an analysis of our results using Cq <36 as the positive cutoff, so this no longer needs to be considered a limitation.However, we have now added the lack of a sample size calculation as a limitation.

Line 370: One of the limitations of this study is the use of retrospective frozen samples with a limited number of samples for each serotype, and we didn’t perfom a sample size calculation to justify this number.

15:

It is difficult to provide labour costs as they can vary significantly between settingsHowever, we have included the cost for the PCR assays and compare the duration of both protocols.

Line 376: The results obtained in this study, using Luna RT-qPCR for direct detection of DENV in patient sera, are promising and could considerably reduce the cost of the PCR assay (from around 14 USD to 4 US per sample), as well as the time to perform the assay (and so the labour cost) by more than 2 (from around 4 hours to less than 2 hours), making it more affordable for diagnosis of dengue in LMICs

16:

We have reviewed the entire manuscript to improve the English and hope it is now acceptable.

Reviewer #2:

1:

Our study already include samples positives for the four serotypes.

2:

This would be indeed interesting but we built our experiment to assess the technical efficiency of the direct RT-qPCR, cost effectiveness evaluation would need additional data and analysis which is not possible to do retrospectively.

3:

The revised manuscript includes this information.

---

## [Decision Letter · Decision Letter 1]

1 Aug 2025

Direct real-time RT-PCR for the detection of dengue virus from patient serum in Lao PDR

PONE-D-25-13860R1

Dear Dr. Dubot-Pérès,

We’re pleased to inform you that your manuscript has been judged scientifically suitable for publication and will be formally accepted for publication once it meets all outstanding technical requirements.

Kind regards,

Nilanka Perera, MD, PhD

Academic Editor

PLOS ONE

Reviewers' comments:

Reviewer's Responses to Questions

**Comments to the Author**

Reviewer #1: All comments have been addressed

Reviewer #2: All comments have been addressed

2. Is the manuscript technically sound, and do the data support the conclusions?

Reviewer #1: Yes

Reviewer #2: (No Response)

3. Has the statistical analysis been performed appropriately and rigorously?

Reviewer #1: Yes

Reviewer #2: (No Response)

4. Have the authors made all data underlying the findings in their manuscript fully available?

Reviewer #1: Yes

Reviewer #2: (No Response)

5. Is the manuscript presented in an intelligible fashion and written in standard English?

Reviewer #1: Yes

Reviewer #2: (No Response)

Reviewer #1: (No Response)

Reviewer #2: (No Response)

**Do you want your identity to be public for this peer review?** For information about this choice, including consent withdrawal, please see our Privacy Policy

Reviewer #1: No

Reviewer #2: No

---

## [Editor Report · Acceptance letter]

PONE-D-25-13860R1

PLOS ONE

Dear Dr. Dubot-Pérès,

I'm pleased to inform you that your manuscript has been deemed suitable for publication in PLOS ONE. Congratulations! Your manuscript is now being handed over to our production team.

Kind regards,

on behalf of

Dr. Nilanka Perera

Academic Editor

PLOS ONE